# Examination of a first-in-class bis-dialkylnorspermidine-terphenyl antibiotic in topical formulation against mono and polymicrobial biofilms

Mariël Miller[1,2]☯, Jeffery C. Rogers[1,2]☯, Marissa A. Badham[1,2], Lousili Cadenas[1,2], Eian Brightwell[1,2], Jacob Adams[1,2], Cole Tyler[1,2], Paul R. Sebahar[3,4], Travis J. Haussener[3,4], Hariprasada Reddy Kanna Reddy[ID][3,4], Ryan E. Looper[3,4], Dustin L. Williams[ID][1,2,3,5,6,7]☯ *

1 George E. Wahlen Department of Veterans Affairs Medical Center, Salt Lake City, UT, United States of America, 2 Department of Orthopaedics, University of Utah, Salt Lake City, UT, United States of America, 3 Curza Global, LLC Provo, UT, United States of America, 4 Department of Chemistry, University of Utah, Salt Lake City, UT, United States of America, 5 Department of Pathology, University of Utah, Salt Lake City, UT, United States of America, 6 Department of Bioengineering, University of Utah, Salt Lake City, UT, United States of America, 7 Department of Physical Medicine and Rehabilitation, Uniformed Services University, Bethesda, MD, United States of America

☯ These authors contributed equally to this work.
* dustin.williams@utah.edu

## Abstract

Biofilm-impaired tissue is a significant factor in chronic wounds such as diabetic foot ulcers. Most, if not all, anti-biotics in clinical use have been optimized against planktonic phenotypes. In this study, an *in vitro* assessment was performed to determine the potential efficacy of a first-in-class series of antibiofilm antibiotics and compare outcomes to current clinical standards of care. The agent, CZ-01179, was formulated into a hydrogel and tested against mature biofilms of a clinical isolate of methicillin-resistant *Staphylococcus aureus* and *Pseudomonas aeruginosa* ATCC 27853 using two separate methods. In the first method, biofilms were grown on cellulose discs on an agar surface. Topical agents were spread on gauze and placed over the biofilms for 24 h. Biofilms were quantified and imaged with confocal and scanning electron microscopy. In the second method, biofilms were grown on bioabsorbable collagen coupons in a modified CDC biofilm reactor. Coupons were immersed in treatment for 24 h. The first method was limited in its ability to assess efficacy. Efficacy profiles against biofilms grown on collagen were more definitive, with CZ-01179 gel eradicating well-established biofilms to a greater degree compared to clinical standards. In conclusion, CZ-01179 may be a promising topical agent that targets the biofilm phenotype. Pre-clinical work is currently being performed to determine the translatable potential of CZ-01179 gel.

**Data Availability Statement:** All relevant data are within the manuscript and its Supporting Information files.

**Funding:** DLW received a Merit Review award from the Department of Veterans Affairs that supported this work. Award number is 1I01RX002287-01. The grant submission was peer reviewed, but the funders had no role in study design, data collection and analysis, decision to publish, or preparation of the manuscript. The funder (Curza Global) provided support in the form of salaries for authors (REL, PRS, HRKR, JCR, TJH, DLW), but did not have any additional role in the study design, data collection and analysis, decision to publish, or preparation of the manuscript. The specific roles of these authors are articulated in the 'author contributions' section. Authors DLW, PRS, REL and TJH have financial interest in Curza Global, LLC.

**Competing interests:** I have read the journal's policy and the authors of this manuscript have the following competing interests: There are multiple patents secured by Curza Global on which multiple authors (DLW, REL, PRS, TJH, HRKR) are listed as inventors. This does not alter our adherence to PLOS ONE policies on sharing data and materials.

## Introduction

The Centers for Disease Control and Prevention (CDC) label the rapid global growth of drug-resistant pathogens "one of our most serious health threats" [1]. The World Health Organization (WHO) also warns that "without urgent, coordinated action by many stakeholders, the world is headed for a post-antibiotic era, in which common infections and minor injuries which have been treatable for decades can once again kill" [2]. Despite this global public health need, the pipeline for new antibiotics, in particular those that display activity against biofilms, is thin [3, 4].

In 2008 the interagency Antimicrobial Availability Task Force, overseen by the Infectious Disease Society of America (IDSA), identified *Pseudomonas aeruginosa* and methicillin-resistant *Staphylococcus aureus* (MRSA), along with four other pathogens, as essential targets to combat antibacterial resistance; these are "ESKAPE" pathogens [4–6]. ESKAPE pathogens represent the paradigms of pathogenesis, transmission, and resistance, and as such, the National Institute of Allergy and Infectious Disease (NIAID) and the IDSA, in coordination with other health organizations, catalyzed initiatives for drug development and research into gene transfer and resistance [2, 7, 8]. These initiatives and promises of funding focus heavily on planktonic-based outcomes, leaving clinicians with minimal alternative options to treat and prevent biofilm-impaired wounds; current selections still constitute topical therapies from the first and second world wars, such as Dakin's solution and colistin, as treatments for chronic wounds [4, 9–11]. Since 2008 only 7 drugs have been approved by the FDA for treatment of "ESKAPE" pathogens, and none are for treatment of *P. aeruginosa* [12–22].

Multiple studies show *S. aureus* to be the most frequent organism responsible for chronic wounds [23–27], but recent studies of chronic wound bacterial profiles provide further evidence supporting the necessity for broad spectrum antibiotic topicals for effective chronic wound therapy [28–30].

An 8-week study at the Wound Healing Center in Copenhagen investigated the bacterial profile of chronic leg ulcers [28]. During the monitoring period, 2 or more bacterial species were identified in 94.4% of the wounds, 4–6 bacterial species were present in 50% of the wounds, and 39% of wounds had more than 6 bacterial species present. The most abundant isolate identified was *S. aureus*, being present in 93.5% of wounds, while *P. aeruginosa* was present in 52.2% of wounds. The abundance of bacterial species per wound can contribute to antimicrobial resistance, and more complex polymicrobial biofilms, making treatment less effective.

These results are supported by studies performed with molecular analysis and imaging on chronic and acute wounds from 123 patients [29, 30]. Biofilm prevalence is similar to other wounds studied, with the majority of chronic wounds containing an abundance of diverse biofilms, while acute wounds have few and less diverse biofilms [28, 29]. Molecular analysis provides greater detail of biofilm ecology within chronic wounds, including the presence of strictly anaerobic bacteria, not seen in growth cultures [29, 30].

Topical therapies are valuable as practitioners consider the inherent characteristics of biofilms. Biofilms are well known to be impervious to systemic antibiotic treatments due in part to reduced activity against persister cells in the anaerobic core of the community [31–35]. In addition, achieving sufficient serum concentrations to eradicate biofilms is often impossible [35]. These clinical paradigms and global scenarios warrant two areas of focus: 1) Development of topical antimicrobial technologies that target and eradicate biofilms. Topical products provide local, high doses of antibiotics that can be applied regularly to sustain antimicrobial delivery. Topical delivery also helps maintain a moist wound bed, which facilitates the prevention of tissue dehydration, accelerates angiogenesis, assists in the breakdown of necrotic tissue and/or

fibrin, and provides for the transport of cytokines and growth factors [36, 37]. 2) Development of novel antimicrobial agents that address the current global threat of antibiotic resistance.

We tested the *in vitro* efficacy of a topical formulation, the active component of which is a compound synthesized as part of a first-in-class series of antibiofilm agents (referred to as CZ compounds). More specifically, CZs are designed and synthesized to specifically eradicate, and in some cases disperse, biofilms via non-specific, global, and rapid disruption of Gram-positive and -negative bacteria [38]. CZs, like other polyamine based antibiotics [39], are considered to exert their effect via membrane permeability and disruption of the lipopolysaccharide (LPS) layer. CZs are synthesized by a straightforward and scalable approach and formulate well with polymer and other carrier agents. We formulated CZ-01179 in a gel and tested its ability to eradicate well-established biofilms of MRSA and *P. aeruginosa*.

Experiments were performed using two separate methods. First, biofilms were grown on cellulose discs following a previously established method by Hammond et al. [40]. Second, biofilms were grown on bioabsorbable collagen in a modified CDC biofilm reactor (Fig 1), then exposed to topical products in a multi-well plate system. We hypothesized that CZ-01179 formulated in a carrier gel would have greater efficacy against biofilms of MRSA and *P. aeruginosa* in mono-microbial and polymicrobial phenotypes than antibiotic-based clinical standards of care.

## Materials and methods

### Bacterial strains

A clinical isolate of MRSA with known pathogenicity was used; it was isolated from a knee-related infection in a patient and produced positive signals of infection in multiple animal models [42, 43]. *P. aeruginosa* ATCC 27853 was purchased from the American Type Culture Collection (ATCC). Each organism was passaged and maintained in Luria-Bertani (LB) broth or on Columbia blood agar at 37˚ C prior to experimentation.

### Supplies and reagents

General supplies, reagents, and growth media were purchased from Fisher Scientific (Hampton, NH). An 8-ply 100% cotton gauze was purchased from Kendall Curity®; Coviden (Mansfield, MA). Five clinically-relevant topical products were purchased via the pharmacy at the Department of Veterans Affairs in Salt Lake City: gentamicin sulfate ointment USP, 0.1% (Perrigo Company, Allegan, MI), mupirocin ointment USP, 2% (Glenmark Pharmaceuticals, Mahwah, NJ), silver sulfadiazine cream, USP 1% (Ascend Laboratories, Montvale, NJ), Neosporin® (400 U Bacitracin Zinc– 3.5mg Neomycin Sulfate– 5,000 U Polymixin B Sulfate; Johnson and Johnson, New Brunswick, NJ), Altabax® (retapamulin ointment) 1%, (GlaxoSmithKline, Barnard Castle, County Durham, United Kingdom). Hyaluronic acid (HA; 1.01 MDa– 1.8 MDa) was purchased from Lifecore Biomedical (Chaska, MN; catalog #HA15M-5). This HA is a bacterial fermentation product of *Streptococcus pyogenes*. Reagents and chemicals for synthesizing CZ-01179 were purchased from Sigma Aldrich (St. Louis, MO). Cellulose discs (6 mm) were purchased from BD (Sparks, MD), and collagen coupons were cut from HeliPlug® Collagen Wound Dressing (Integra LifeSciences, Plainsboro, NJ). BacLight™ Bacterial Viability kits were purchased from Molecular Probes (Eugene, OR). The Nunc™ Lab-Tek™ chamber slide system was purchased from ThermoScientific™ (Waltham, MA).

### CZ-01179 synthesis and gel formulation

CZ-01179 was synthesized (Fig 2) by the following method: to a stirring solution of a dicarbaldehyde (5'-(tert-butyl)-[1,1':3',1"-terphenyl]-4,4"-dicarbaldehyde: 2.12 g, 6.22 mmol, 1 equiv.)

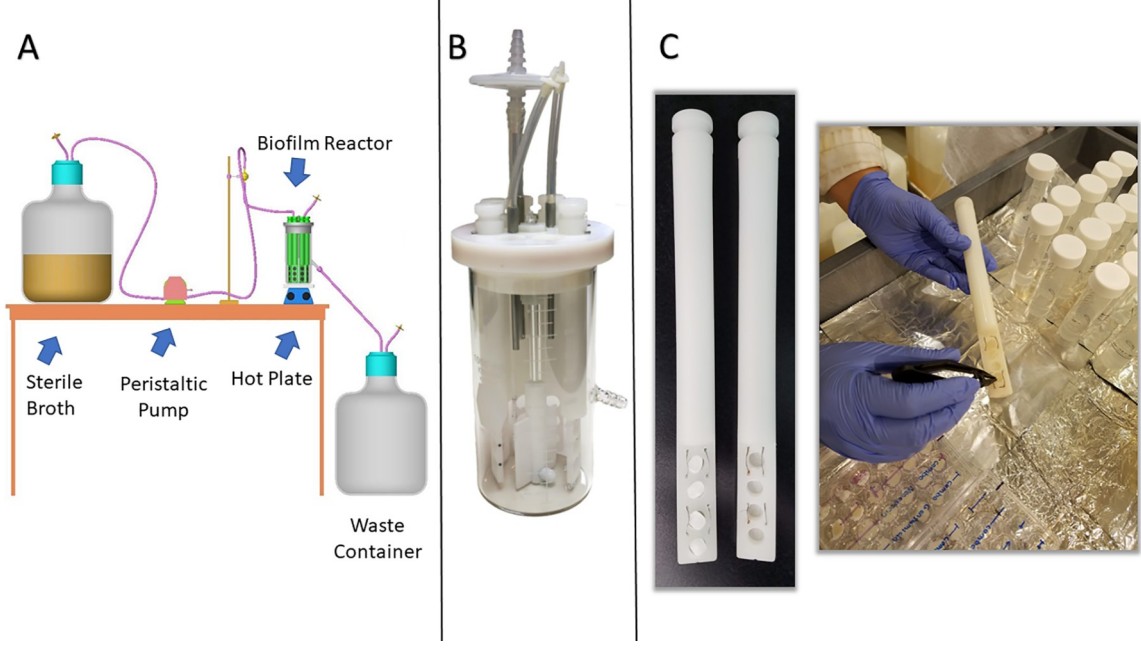

**Fig 1. Setup of a modified CDC biofilm reactor for growing biofilms on collagen [41].** (A) Schematic of how a CDC biofilm reactor is set up in general. Source: BioSurface Technologies (B) Image of a fully assembled CDC biofilm reactor. (C) Reactor rods were modified to hold collagen coupons (left panel). A snapshot of the process is shown for removing a collagen coupon on which mature biofilm was grown (right panel). Coupons were rinsed in conical tubes, then used for subsequent analysis. This figure was reused from Williams et al. [41] under the Creative Commons Attribute (CC BY) license.

in MeOH (100 mL) and DCE (25 mL) at 0°C was added the diamine (N1-(3-aminopropyl)-N3-(2-ethylbutyl)propane-1,3-diamine: 3.61 g, 16.8 mmol, 2.7 equiv.) portion-wise over 20 min. The solution was stirred for 16 h. $NaBH_4$ (0.95 g, 24.9, 1 equiv.) was added portion-wise over 20 min and the reaction stirred for an additional 1 h. The solvent was evaporated, and the crude solid partitioned between EtOAc (500 ml) and 10% NaOH (250 ml). The NaOH phase was washed with EtOAc (500 ml), and the combined organics were dried over $Na_2SO_4$. Column chromatography was performed using gradient conditions starting at (300:16:1 $CH_2Cl_2$: $MeOH:NH_4OH$). The free base was acidified with HCl in MeOH (100 ml) and cooled to 0°C for 1 h. The resulting precipitate was filtered and dried to afford the HCl salt as a white solid (25–52%). Recrystallization with $H_2O$ (solvent) and *i*PrOH (anti-solvent) delivered analytically pure material. $^1H$ NMR (500 MHz, $D_2O$) δ ppm 7.78–7.69 (m, 7H), 7.61 (bs, 4H), 4.38 (s, 4H), 3.26–3.20 (m, 16H), 3.01 (s, 4H), 2.17 (bs, 8H), 1.67 (bs, 2H), 1.38 (bs, 17H), 0.88 (s, 12H). $^{13}C$

**Fig 2. Schematic of CZ-01179 synthesis and resultant compound structure.**

NMR (125 MHz, $D_2O$) δ ppm 153.4, 141.8, 140.4, 130.4, 129.7, 127.8, 123.8, 122.9, 50.9, 50.9, 44.9, 44.6, 43.9, 37.6, 34.4, 30.5, 22.6, 22.4, 22.4, 9.4. IR (neat): 3334 (bs), 2963, 2766, 1457 (all s) cm$^{-1}$. mp decomposition (180–184˚C). LRMS Calculated for $C_{48}H_{80}N_6$ m/z 741.6 $[M+H]^+$, Obsd. 370.7 $[M+H]^+$/2.

CZ-01179 was formulated into a gel following synthesis. CZ-01179 powder was added to sterile PBS; three separate solutions were made with final concentrations of 0.5% (5 mg/mL), 1% (10 mg/mL) or 2% (20 mg/mL). After the CZ-01179 dissolved, HA was added to a final concentration of 1.5% (15 mg/mL). Each gel had the same HA concentration. The gel was mixed thoroughly and allowed to settle for a minimum of 24 h at room temperature before being used for experimentation. At a 2% concentration, pH of the gel was approximately 7.0. Gels with CZ-01179 at 0.5% and 1.0% were initially at pH 4 and were adjusted with NaOH to a pH of approximately 7.0 prior to experimentation. The pH of clinically-relevant gels was not adjusted so as to reflect clinical state.

## Cytotoxicity

A minimal essential media (MEM) elution assay was conducted by Nelson Laboratories to determine cytotoxicity profiles (ISO 10993–5) of the 2% CZ-01179 gel and clinical standards. Test articles and controls were extracted in 1x minimal essential media (MEM) with 5% bovine serum for 24–25 h at 37 ± 1˚C with agitation. Multiple well cell culture plates were seeded with a verified quantity of industry standard L-929 cells (ATCC CCL-1) and incubated until ~80% confluent. The test articles were held at room temperature for less than four h before testing. The extract fluids were not filtered, centrifuged or manipulated in any way following the extraction process. The test extracts were added to the cell monolayers in triplicate. The cells were incubated at 37 ± 1˚C with 5 ± 1% CO2 for 48 ± 3 h.

Cell monolayers were examined and scored (0–4) based on the degree of cellular destruction. Specifically, Grade 0 = No reactivity, no cell lysis; Grade 1 = Slight reactivity, ≤20% rounding, occasional lysis; Grade 2 = Mild reactivity, 20% ≤ 50% rounding, no extensive cell lysis; Grade 3 = Moderate reactivity, 50% ≤ 70% rounding and lysed cells; Grade 4 = Severe reactivity, nearly complete destruction of cell layers. Testing was performed in compliance with US FDA goods and manufacturing practice (GMP) regulations 21 CFR Parts 210, 211 and 820.

## Cellulose disc assay

A modified protocol of Hammond et al. was used to grow biofilms on cellulose discs and test efficacy of topical antibiotic products [40]. Isolates were grown in LB broth overnight (~24 h) at 37˚C. A 1 mL aliquot was placed into a microcentrifuge tube, pelleted at 12,000 rpm for 5 min, washed, and resuspended in 1 mL of fresh LB as a stock culture. A baseline of colony forming units (CFU)/mL in stock cultures was determined for each experiment using a 10-fold dilution series. The stock concentration of MRSA was ~3.6 x 10$^9$ CFU/mL, and for *P. aeruginosa* was ~4.7 x 10$^9$ CFU/mL.

An n = 8 cellulose discs were sterilely and equidistantly placed on the surface of a single LB agar plate. Fifty μL of bacterial suspension were pipetted onto the surface of each disc. Only one isolate was inoculated/plate so as not to have cross contamination. Plates were incubated at 37˚C for 48 h to allow biofilms to form on the cellulose material. Notably, this incubation time differed from the Hammond et al. method, which suggested an incubation time of 24 h [40]. The rationale for a longer incubation period was to form more robust biofilms; minimal and immature biofilms were observed by scanning electron microscopy (SEM) with 24 h

growth. A more significant bioburden challenge was desired for topical assessment, thus biofilms were grown for 48 h.

Approximately 800 mg of topical agent were spread in a thin layer, i.e., "buttered" on sterile 2" x 2" cotton gauze. The "buttered" side of the gauze pad was placed in contact with the discs such that all n = 8 discs were covered completely. Three additional gauze pads were placed on top of the "buttered" gauze pad; the rationale was to increase the gauze height so that the lid of the Petri dish compressed the stack and simulated pressure of a bandage over a wound (recommended by Hammond et al.) [40]. Masking tape held the Petri dish lid in place and it was incubated for 24 h at 37°C. Biofilms of MRSA were treated with CZ-01179, mupirocin, gentamicin, silver sulfadiazine, retapamulin, or Neosporin®. Biofilms of *P. aeruginosa* were treated with CZ-01179, gentamicin, silver sulfadiazine, or Neosporin® (retapamulin and mupirocin are indicated for Gram-positive organisms).

Cellulose discs were sterilely removed and placed individually into 1 mL of PBS. Samples were vortexed for 1 min, sonicated for 10 min at 42 kHz and plated using a 10-fold dilution series to quantify the CFU/disc that remained after treatment. Positive controls of growth (n = 8) with no treatment were also quantified for comparison.

The same growth protocol as outlined above was used to test efficacy of topical products against polymicrobial biofilms. However, inocula concentrations were varied to grow MRSA and *P. aeruginosa* as polymicrobial biofilms. When the two isolates were inoculated at 1:1 or even 1:1,000 ratio, *P. aeruginosa* overwhelmed the MRSA isolate. As such, a 1:10,000 ratio was used; MRSA was inoculated at a concentration 10,000 times higher than *P. aeruginosa*. Each isolate was suspended to a turbidity of 10% using a nephelometer (concentration equated to ~1 x $10^9$ CFU/mL). MRSA was diluted 1:1,000 (~1 x $10^6$ CFU/mL) and *P. aeruginosa* was diluted 1:10,000,000 (~1 x $10^2$ CFU/mL) using a 10-fold dilution series. Twenty-five µL of each solution were pipetted onto cellulose discs for a total of 50 µL per sample. Polymicrobial biofilm growth was quantified as described above to obtain a baseline of CFU/disc.

Biofilms were observed qualitatively on cellulose discs using confocal laser scanning microscopy (CLSM); a BacLight™ Bacterial Viability kit was used in low light. Treated and untreated cellulose discs (n = 8 from each test) were removed from the agar, stained and fixed in a Nunc™ Lab-Tek™ chamber slide system. Using the chamber slide allowed for staining and fixation all in one chamber so as to preserve the structure of the biofilm prior to evaluation with CLSM. Manufacturer instructions were followed: 3 µL of SYTO 9 3.34 mM and 3 µL of propidium iodide 20 mM per 1 mL of sterile water. A 100 µL volume submerged each cellulose disc in an individual chamber. Chamber slides were covered in aluminum foil to shield from light for 15 min in order for the stain to integrate with the bacterial cells. Stain was decanted from each chamber and samples were washed with sterile PBS. Each sample was fixed with 100 µL of 10% neutral buffered formalin for 30 min, then washed again with 100 µL of PBS. Samples were air dried in the same dark environment and viewed by CLSM.

SEM images were also collected to observe surface morphologies of monomicrobial and polymicrobial growth on cellulose discs. Separate cellulose discs (not used in Live/Dead assays) were fixed for a minimum of 2 h in modified Karnovsky's fixative (2.5% glutaraldehyde, 2% paraformaldehyde in PBS buffered to pH 7.2–7.4). Samples were dehydrated in 100% ethanol for at least 2 h, air dried, sputter coated with gold, then imaged in a JEOL JSM-6610 SEM.

## Collagen coupon assay

Monomicrobial and polymicrobial biofilms of MRSA and *P. aeruginosa* were also grown on bioabsorbable collagen and exposed to topical therapies. The rationale was two-fold: 1) to

perform experiments using a material that would more closely model a physiological substrate, and 2) as will be shown, the Hammond et al. method had important limitations that led to inconsistent results—motivating a secondary analysis [40].

Biofilms were grown on collagen using a modified CDC biofilm reactor (Fig 1A and 1B) as described previously [41]. Rather than using standard coupon rods, blank polypropylene rods were purchased and 4 holes (8 mm diameter) were drilled into the bottom portion. The holes were drilled half way through the rod (Fig 1C). HeliPlug™ collagen coupons were sterilely cut to size (5 mm x 10 mm), and pressed into each bored-out cavity of a rod (Fig 1C). Assembly was performed in a biosafety cabinet to maintain sterility.

Five-hundred mL of brain heart infusion (BHI) broth were added to the reactor after it was assembled. The broth was aseptically inoculated with $10^5$ CFU/mL (adjusted from 0.5 McFarland standard) of MRSA or *P. aeruginosa* for monomicrobial biofilm growth. The reactor was set on a hot plate at 34˚C and a baffle rotation of 130 rpm. Bacteria were grown in batch phase for 24 h, after which a 10% solution of BHI was flowed through the reactor at a rate of 6.94 mL/min using a peristaltic pump (MasterFlex L/S Microbore, Cole Palmer, Vernon Hills, IL) for an additional 24 h (Fig 1A).

The inoculation protocol to grow polymicrobial biofilms on collagen was similar to cellulose; each isolate was suspended to a 0.5 McFarland Standard (~1 x $10^8$ CFU/mL). MRSA was diluted 1:1,000 (~1 x $10^5$ CFU/mL) and *P. aeruginosa* was diluted 1:10,000,000 (~1 x 10 CFU/mL). However, we determined experimentally that polymicrobial biofilms grew more successfully if the reactor was inoculated multiple times with bacterial solution (one isolate would typically outcompete the other with a single inoculation). The reactor was inoculated five times total: at setup, then 15 min, 45 min, 1 h 45 min, and 3 h 45 min after the initial inoculation. Polymicrobial biofilms were otherwise grown as described for monomicrobial growth.

The efficacy of each topical product was determined against biofilms on collagen by placing 1g of product into a well of a 24-well plate. Collagen coupons were sterilely removed from the reactor and placed individually atop the topical product. Each coupon was then covered with an additional 1g of product, and the lid of the plate replaced. This procedure exposed biofilms on all surfaces of the collagen to a topical product, which was an important difference compared to the Hammond et al. method [40]; in the Hammond et al. method, only the top side of a cellulose disc was exposed to topical treatment, with the underside remaining untreated against the agar surface.

Samples were incubated with their respective topical agents (same as tested against cellulose discs) for 24 h at 37˚C. Collagen coupons were removed, rinsed 3x in PBS and placed into 2 mL of PBS. Each sample was vortexed for 1 min, sonicated at 42 kHz for 10 min, vortexed again and plated in triplicate on trypticase soy agar (TSA) using a 10-fold dilution series. Selective agar was made to resolve growth between *S. aureus* and *P. aeruginosa*; TSA + triclosan 0.625 μg/mL for *P. aeruginosa*, and TSA + 7.5% NaCl for MRSA. Selective agar protocols were adapted from previous methods and confirmed with polymicrobial biofilm and planktonic quantification [44–47]. Agar plates were incubated at 37˚C and colonies counted at 24 and 48 h. Positive controls of growth (not exposed to antibiotic treatments) were quantified for baseline comparison. All data were collected with n = 8 repeats.

## Statistical analysis

Results were analyzed using a one-way ANOVA with Tukey post hoc analysis and alpha level at 0.05 in SPSS software (IBM Corp., Armonk, NY).

## Results

### Cytotoxicity

MEM elution tests showed that CZ-01179 had equivalent cytotoxicity outcomes compared to clinical topical products. Mupirocin, gentamicin, retapamulin, and silver sulfadiazine received failing scores; all n = 3 samples of clinically-relevant topicals scored 4 on a scale of 0–4 (score of 3–4 being considered failure and 0–2 considered passing). All n = 3 samples of CZ-01179 also had a score of 4. Neosporin® was the only topical product to receive passing scores of 1, 2, and 1 for the three samples tested.

### Cellulose disc assay

The 48-h biofilm growth protocol for cellulose produced well-established, mature monomicrobial and polymicrobial biofilms of both MRSA and *P. aeruginosa* (Fig 3A–3C). However, quantification outcomes following efficacy analyses were highly variable, in particular with the clinically-relevant products. We established a sub-hypothesis after observing the inconsistent outcomes: we hypothesized that topical treatments failed to reach the biofilms that formed on the underside of the cellulose disc (immediately adjacent to the surface of the agar), and the lack of exposure in that region led to highly variable quantification data.

To test the sub-hypothesis, biofilms were grown on cellulose following the same growth protocol outlined above. SEM and CLSM imaging was performed to determine: 1) if biofilms formed on the underside of the cellulose fiber network that was in apposition to the agar surface, and 2) if those biofilms on the underside of cellulose discs were still viable following the topical product delivery protocol.

SEM imaging confirmed the presence of biofilms on, within, and between the interstices of the fibers on the underside of cellulose discs (Figs 3 and 4). Live/Dead imaging also indicated

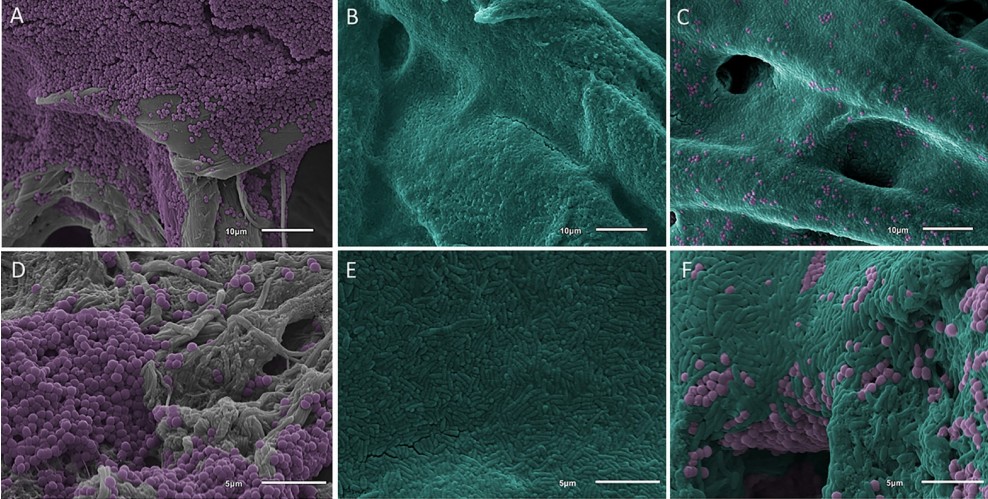

**Fig 3. SEM images of MRSA (colored purple) and *P. aeruginosa* (colored teal) biofilms grown on cellulose and collagen.** (A) MRSA biofilms on cellulose (gray substrate) after 48 h of growth. (B) *P. aeruginosa* biofilms on cellulose after 48 h of growth. The sheet-like growth of *P. aeruginosa* biofilms covered the substrate completely. (C) Polymicrobial biofilms of MRSA and *P. aeruginosa* on cellulose after 48 h of growth. *P. aeruginosa* grew in sheet-like structures with MRSA clusters observed throughout. (D) MRSA biofilm on collagen (gray substrate) after 48 h of growth. (E) *P. aeruginosa* biofilm on collagen after 48 h of growth. (F) Polymicrobial biofilms on collagen after 48 h of growth. Morphology was similar to that of cellulose with *P. aeruginosa* growing in sheet-like structures with MRSA clusters interspersed throughout.

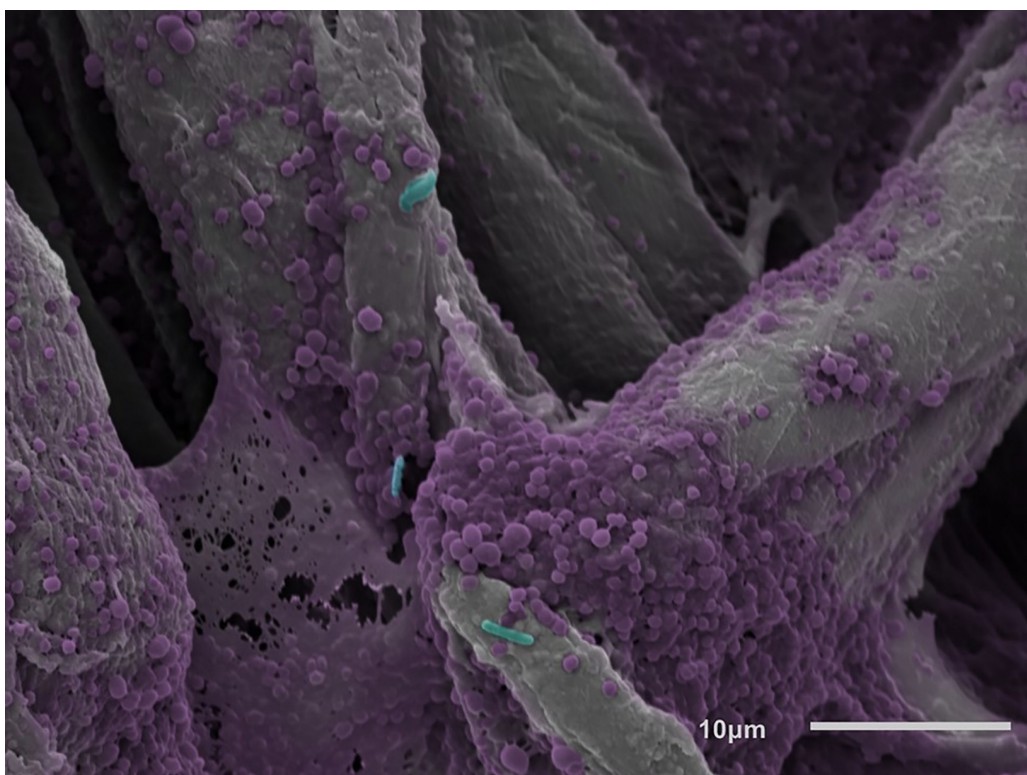

**Fig 4. Polymicrobial biofilm on the non-treated side of a cellulose disc.** Biofilm growth can be seen within chasms and voids between cellulose fibers.

that biofilms were viable on all surfaces of cellulose discs, but only surfaces in direct contact with topical agents showed cell death; bacteria on the underside and center of cellulose discs stained green (living), supporting our sub-hypothesis that bacteria on untreated surfaces were still viable and were not exposed to topical product treatments (Figs 5 and 6) [41].

Despite the limitation of this method, Live/Dead staining provided some useful information on topical efficacy. Confocal imaging and staining indicated that CZ-01179 was highly effective against well-established biofilms that were exposed to the formulated gel, whereas clinical products had limited efficacy (Fig 6). These outcomes provided rationale for performing analysis on collagen coupons.

## Collagen coupon assay

Biofilms on collagen grew to maturity (Fig 3), and SEM images indicated more robust biofilm formation compared to cellulose discs. Quantification of positive controls supported this observation with ~1 $\log_{10}$ more CFU/coupon compared to cellulose discs for both isolates.

Quantification data from efficacy testing against biofilms on collagen are reported in Table 1. Outcomes indicated that of the clinical standards of care, gentamicin was most effective against both monomicrobial and polymicrobial biofilms of MRSA and of *P. aeruginosa* (Fig 7 and Table 1). Gentamicin showed a $\log_{10}$ reduction of 3.56 CFU/collagen in monomicrobial biofilms of MRSA, and against polymicrobial biofilms it was effective against MRSA with a $\log_{10}$ reduction of 5.21 CFU/collagen. Against both monomicrobial and polymicrobial biofilms, gentamicin showed complete eradication of *P. aeruginosa*, with no detectable growth

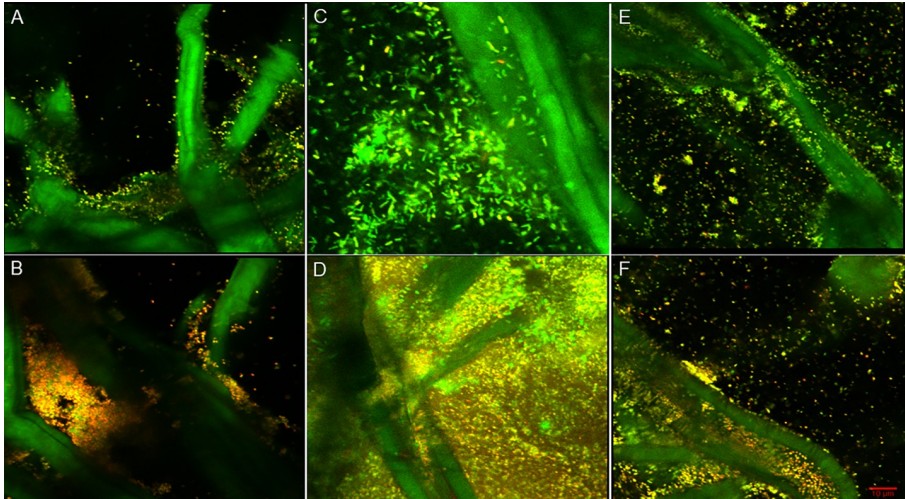

**Fig 5. Single section CLSM (60x magnification) images collected using BacLight™ Live/Dead stain to determine whether CZ-01179 (2%) affected biofilms on the bottom as well as top of portions of cellulose discs.** Cellulose fibers stained green along with live cells. (A) MRSA biofilm on the bottom (non-treated) side of a cellulose disc. The majority of cells stained green (living), suggesting limited activity against the biofilms in that region. (B) MRSA biofilm on the top (treated) side of a cellulose disc. The predominance of red indicated that there was significant antimicrobial activity against biofilms in that region. (C) *P. aeruginosa* biofilm on the bottom (non-treated) side of a cellulose disc. The predominance of green stain suggested very little, if any activity was present against the biofilms. (D) Biofilm of *P. aeruginosa* on the top (treated) side of a cellulose disc. The majority of cells stained red, suggesting significant antimicrobial activity, in particular compared to the untreated side. (E) Polymicrobial biofilms on the bottom (non-treated) side of a cellulose disc. The lack of red/yellow indicated little to no antimicrobial activity had occurred. (F) Polymicrobial biofilms on the top (treated) side of a disc. The majority of cells stained red, suggesting that CZ-01179 was effective at eradicating polymicrobial biofilms on the top portions of cellulose discs.

(Table 1). At all three concentrations (0.5%, 1%, 2%) CZ-01179 reduced all monomicrobial and polymicrobial biofilms to below detectable levels (Fig 7 and Table 1; p = 0.001 or less in all cases compared to controls).

Mupirocin and retapamulin were only tested against monomicrobial biofilms of MRSA, as these topicals are FDA approved for staphylococcal species. Mupirocin-treated MRSA biofilms showed a $\log_{10}$ reduction of 3.17 CFU/collagen, indicating similar efficacy as gentamicin against MRSA (Table 1). Retapamulin-treated MRSA biofilms had a $\log_{10}$ reduction of 1.62 CFU/collagen (Table 1).

Data were compared statistically using an independent samples *t* test with alpha set at 0.05. Tests were run with n = 8 samples/group.

Silver sulfadiazine and Neosporin® are sold as broad spectrum antimicrobial topicals, and were tested against monomicrobial and polymicrobial biofilms of MRSA and *P. aeruginosa*. Against monomicrobial biofilms of MRSA, collagen treated with silver sulfadiazine had a $\log_{10}$ reduction of 2.97, while collagen treated with Neosporin® had a $\log_{10}$ reduction of 2.46 (Table 1). Similar efficacy was seen when applied to polymicrobial biofilms with MRSA $\log_{10}$ reductions of 4.20 and 2.42 CFU/collagen (Table 1), respectively. Both topicals were less effective against *P. aeruginosa* in monomicrobial biofilms, with 1.55 and 0.87 $\log_{10}$ reductions, respectively. The same was observed for polymicrobial biofilms; silver sulfadiazine showed 3.74 $\log_{10}$ reductions and Neosporin® showed no reduction (Table 1 and Fig 7). These data are consistent with current literature on the efficacy of Neosporin® and silver sulfadiazine [48].

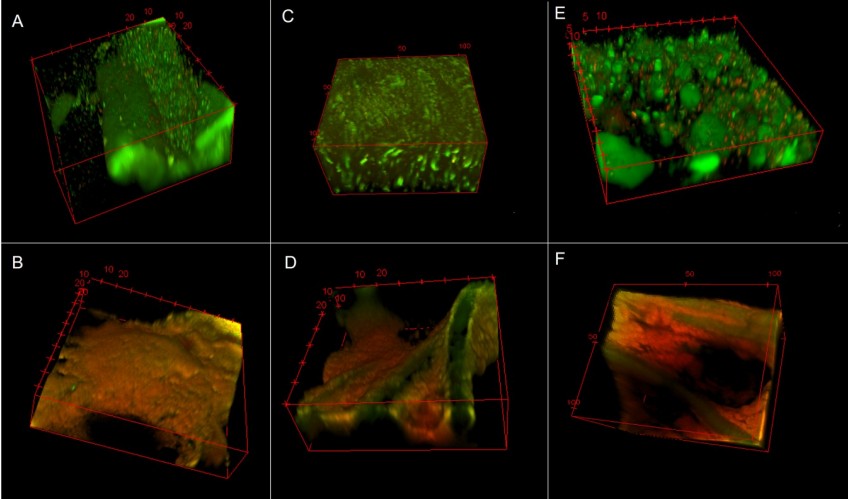

**Fig 6. 3D reconstructed CLSM images showing effect of representative topical agents against biofilms on the top (treated) side of cellulose discs.** Images were collected with BacLight™ Live/Dead stain. (A) MRSA biofilm treated with gentamicin (0.1%). The predominance of green (living cells) indicated there was limited antimicrobial activity against well-established biofilms. (B) MRSA biofilm treated with CZ-01179 (2%). The antimicrobial gel was highly effective at eradicating well-established biofilms. (C) *P. aeruginosa* biofilm treated with gentamicin (0.1%). The significant amount of living cells post treatment of gentamicin, demonstrates the limitations of the Hammond et al. method, as *P. aeruginosa* biofilms grown on collagen treated with gentamicin showed complete eradication (43). (D) *P. aeruginosa* biofilm treated with CZ-01179 (2%). The antibiofilm agent was able to disrupt the sheet-like structures of the biofilm. (E) Polymicrobial biofilms treated with silver sulfadiazine showed minimal efficacy. (F) Deep and widespread antimicrobial activity was observed within the matrix of the polymicrobial biofilm treated with CZ-01179 (2%) gel.

**Table 1. Remaining $\log_{10}$ transform CFU/collagen coupon following 24 h of topical treatment.** Each collagen coupon received 2g of topical agent applied in the specified concentrations.

| | | Monomicrobial Biofilms | | | Polymicrobial Biofilms | | |
|---|---|---|---|---|---|---|---|
| | | Average | St Dev | (*p* value) | Average | St Dev | (*p* value) |
| **Positive Control (Baseline Quantification)** | MRSA | 9.37 | 0.30 | | 9.06 | 0.50 | |
| | *P. aeruginosa* | 8.52 | 0.34 | | 7.55 | 0.49 | |
| **Silver Sulfadiazine 1%** | MRSA | 6.40 | 0.34 | (0.001) | 4.86 | 0.53 | (0.001) |
| | *P. aeruginosa* | 6.98 | 1.12 | (0.001) | 3.81 | 1.60 | (0.001) |
| **Gentamicin 0.1%** | MRSA | 5.81 | 0.56 | (0.001) | 3.85 | 0.31 | (0.001) |
| | *P. aeruginosa* | 0.00 | 0.00 | (0.001) | 0.00 | 0.00 | (0.001) |
| **Mupirocin 2%** | MRSA | 6.21 | 0.52 | (0.001) | | | |
| | *P. aeruginosa* | | | | | | |
| **Neosporin®** | MRSA | 6.91 | 0.52 | (0.001) | 6.64 | 0.32 | (0.001) |
| | *P. aeruginosa* | 7.65 | 0.33 | (0.001) | 8.12 | 0.41 | (0.001) |
| **Retapamulin 1%** | MRSA | 7.75 | 0.35 | (0.001) | | | |
| | *P. aeruginosa* | | | | | | |
| **CZ-01179 0.5%** | MRSA | 0.00 | 0.00 | (0.001) | 0.00 | 0.00 | (0.001) |
| | *P. aeruginosa* | 0.00 | 0.00 | (0.001) | 0.00 | 0.00 | (0.001) |
| **CZ-01179 1%** | MRSA | 0.00 | 0.00 | (0.001) | 0.00 | 0.00 | (0.001) |
| | *P. aeruginosa* | 0.00 | 0.00 | (0.001) | 0.00 | 0.00 | (0.001) |
| **CZ-01179 2%** | MRSA | 0.00 | 0.00 | (0.001) | 0.00 | 0.00 | (0.001) |
| | *P. aeruginosa* | 0.00 | 0.00 | (0.001) | 0.00 | 0.00 | (0.001) |

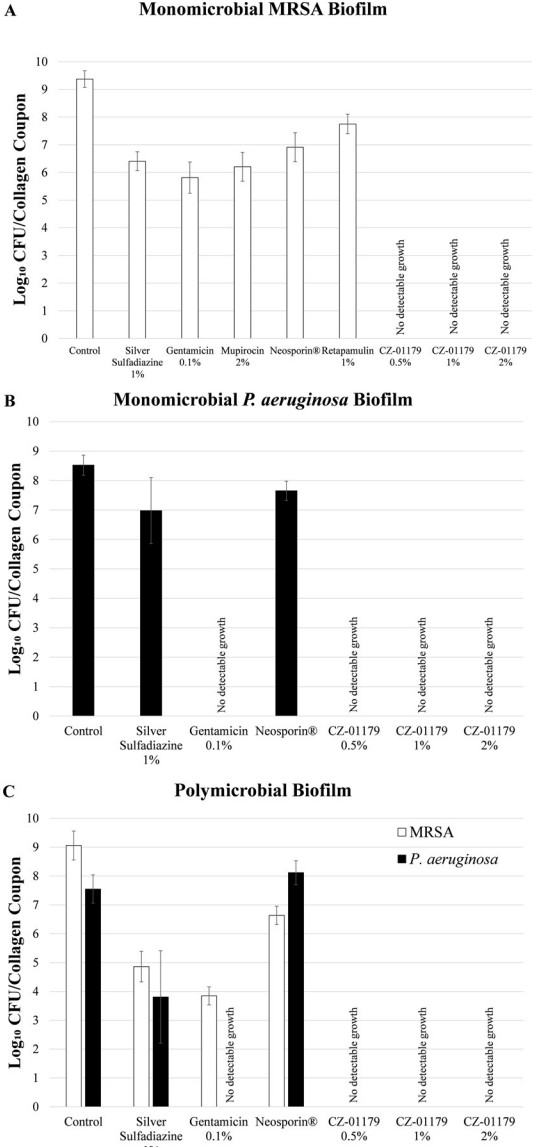

**Fig 7.** (A) Efficacy of topical agents against MRSA biofilms on collagen. (B) Efficacy of topical agents against *P. aeruginosa* biofilms on collagen. (C) Efficacy of topical agents against polymicrobial biofilm growth on collagen.

## Discussion

Chronic wounds affect millions of patients annually. They lead to significant morbidity and burden healthcare systems with cost and complexity. When affected by biofilm-related infection, treatment becomes even more challenging. Quality of life can be affected and may lead to amputation. Developing a topical therapy that targets the biofilm phenotype and reduces rates of infection would be an important advancement in clinical care of these wound types. This study assessed the *in vitro* activity of an antibiofilm gel against MRSA and *P. aeruginosa* in monomicrobial and polymicrobial biofilms in two separate growth systems as part of a translational process toward this objective.

The efficacy profiles of CZ-01179 gel and clinical products were first collected following a method published by Hammond et al. [40]. Biofilm formation throughout cellulose discs was

confirmed by SEM, and CLSM indicated that bacteria on the underside of discs were not eradicated as treatments were only administered to the topside of a disc. Given variable diffusion of topical products this led to variable outcomes, complicating data interpretation. We conclude that this method may not be ideal for assessing efficacy of topical products unless it is modified to control for the lack of exposure to biofilms that are adjacent to the agar surface.

CZ-01179 gels (all three concentrations) had equal efficacy against *P. aeruginosa* biofilms as gentamicin in the collagen test (Fig 7). CZ-01179 gels were more efficacious at eradicating biofilms in all other cases when compared to the standard of care topicals in the collagen tests (Fig 7). Gentamicin had the greatest $\log_{10}$ reduction against monomicrobial biofilms of *P. aeruginosa* and polymicrobial biofilms amongst the clinical standards of care. These data were promising, but broader-scale consideration is given in clinical context; aminoglycosides (including gentamicin) are susceptible to both intrinsic and imported resistance mechanisms by *P. aeruginosa* [10]. These antibacterial resistant strains of *P. aeruginosa* result in ~6,700 infections per year in the USA alone [1]. These and other limitations motivate and warrant development of additional topical products such as CZ gel that are active against Gram-positive and -negative bacteria provides. Additional product options can help relieve the selective pressures leading to multidrug resistant pathogens.

Mupirocin reduced MRSA CFU by 3.17 $\log_{10}$ units on collagen. Mupirocin is a commonly deployed antibiotic against Gram-positive bacteria such as *S. aureus* including MRSA, however, there are some limitations to be considered when targeting biofilms. GlaxoSmithKline explains that mupirocin, under the trade name Bactroban®, has limited activity towards anaerobic organisms [49]. A prominent characteristic of mature biofilms is the oxygen gradient that is present at various levels throughout the structure, resulting in a predominantly anaerobic core [35, 50]. This anaerobic core allows for polymicrobial diversity in biofilms, demonstrated by the presence of several strictly anaerobic bacteria within patient chronic wounds, and unaffected by many standard of care topicals [29]. Because *S. aureus* is a well-known facultative anaerobic organism, it can thrive at all levels along the oxygen gradient in varying metabolic stages. As many standards of care have limited activity on the anaerobic core of a biofilm, the diversity provided by anaerobes and anaerobic phenotypes contributes to prolonged chronicity of biofilm-impaired wounds such as diabetic foot ulcers. Furthermore, like many antibiotics, mupirocin has also been implicated in resistance development by strains of *S. aureus* [51–53]. Prolonged clinical use of mupirocin is generally not recommended, especially in settings of endemic MRSA colonization [54].

The 2 $\log_{10}$ difference in MRSA reduction between monomicrobial and polymicrobial biofilms can likely be attributed to selective pressures of *P. aeruginosa* against *S. aureus*; specifically, excretion of LasA protease and 4-hydroxy-2-heptylquinoline-*N*-oxide (HQNO), known anti-staphylococcal agents [45–47, 55–58]. It is possible that a longer study would show an initial decrease in MRSA CFUs in polymicrobial biofilms, but then a greater resurgence, as the gene giving resistance to HQNO also provides protection against aminoglycosides, including gentamicin [10, 32, 45, 58]. The HQNO-induced *S. aureus* strain is easily identified by its small size and slow growth [45, 56, 58]. Due to natural limitations of *in vitro* studies, the slow growth of this mutated *S. aureus* strain might conceal the population of surviving cells. Current *in vivo* studies with a monitoring period post treatment, are being conducted by our laboratory which will provide a more accurate assessment of CZ efficacy against polymicrobial biofilms.

While our quantification data showed that samples treated with CZ-01179 had no detectable surviving bacteria, Live/Dead staining would provide further evidence for its efficacy. Unfortunately, the porous and fibrous nature of the collagen HeliPlug® make it unsuitable for confocal imaging. Finding an additional method for imaging would help us in confirming topical efficacy and biofilm disruption.

Efficacy data resulted in *p* values that were equivalent across experimental groups when compared against controls (Table 1). While statistical significance is an important parameter for comparing data sets, clinical significance should also be considered. All of the clinically-relevant topical products but gentamicin had a bioburden that remained above $10^5$ CFU. Though challenged, a $10^5$ level is still used as a clinical rule of thumb to indicate a level of bioburden that may cause infection [59]. We seek to develop antibiofilm agents that can effectively reduce biofilm levels to below the $10^5$ level. These *in vitro* experiments indicated that CZ-01179 achieved that, yet *in vivo* data is needed to make a more definitive conclusion. We recently published the first *in vivo* data set of CZ-01179 against biofilms of *Acinetobacter baumannii* and showed greater than a $10^5$ reduction [60]. *In vivo* work is currently ongoing with MRSA and *P. aeruginosa.*

When compared against clinical standards in a MEM elution test, CZ-01179 gel was found to have similar cytotoxicity profiles at therapeutic concentrations. The balance of cytotoxicity is an important consideration, recognizing that infection is also toxic. Promisingly, recent *in vivo* data indicate that cytotoxicity in topical applications are not seen [60].

Antimicrobial delivery by topical therapy provides the ability to achieve high antimicrobial doses necessary for eradication of a biofilm that would otherwise be unachievable with systemic therapy. CZs constitute a promising class of compounds as they have reduced risk of resistance, formulate well with polymers, are highly soluble and stable (beneficial for hydrogel materials), and are not limited in efficacy by the metabolic state of bacteria in a biofilm. Taken together, data indicated that CZ-01179 is a candidate for advancement toward *in vivo* testing to assess its ability to treat and/or prevent biofilm-related wound infections caused by mono-microbial or polymicrobial biofilms.

## Supporting information

**S1 Table. Raw microbiological data for MRSA.**
(PDF)

**S2 Table. Raw microbiological data for *P. aeruginosa* (ATCC 27853).**
(PDF)

**S3 Table. Raw microbiological data for polymicrobial biofilms composed of MRSA and *P. aeruginosa* (ATCC 27853).**
(PDF)

## Acknowledgments

The authors thank Scott Porter for his technical assistance.

## Author Contributions

**Conceptualization:** Paul R. Sebahar, Travis J. Haussener, Hariprasada Reddy Kanna Reddy, Ryan E. Looper, Dustin L. Williams.

**Data curation:** Marissa A. Badham, Lousili Cadenas, Eian Brightwell, Jacob Adams, Cole Tyler, Ryan E. Looper, Dustin L. Williams.

**Formal analysis:** Mariël Miller, Jeffery C. Rogers, Cole Tyler, Paul R. Sebahar, Ryan E. Looper, Dustin L. Williams.

**Funding acquisition:** Dustin L. Williams.

**Investigation:** Eian Brightwell, Ryan E. Looper, Dustin L. Williams.

**Methodology:** Marissa A. Badham, Lousili Cadenas, Eian Brightwell, Jacob Adams, Paul R. Sebahar, Travis J. Haussener, Hariprasada Reddy Kanna Reddy, Ryan E. Looper, Dustin L. Williams.

**Project administration:** Mariël Miller, Dustin L. Williams.

**Resources:** Lousili Cadenas, Travis J. Haussener, Hariprasada Reddy Kanna Reddy, Dustin L. Williams.

**Software:** Cole Tyler.

**Supervision:** Mariël Miller, Jeffery C. Rogers, Ryan E. Looper.

**Validation:** Mariël Miller, Jeffery C. Rogers, Jacob Adams, Dustin L. Williams.

**Writing – original draft:** Dustin L. Williams.

**Writing – review & editing:** Mariël Miller, Jeffery C. Rogers, Marissa A. Badham, Paul R. Sebahar, Ryan E. Looper, Dustin L. Williams.

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
