## [Decision Letter · Decision Letter 0]

25 Jun 2020

PONE-D-20-16178

Examination of a first-in-class bis-dialkylnorspermidine-terphenyl antibiotic in topical formulation against mono and polymicrobial biofilms

PLOS ONE

Dear Dr. Williams,

Thank you for submitting your manuscript to PLOS ONE. After careful consideration, we feel that it has merit but does not fully meet PLOS ONE’s publication criteria as it currently stands. Therefore, we invite you to submit a revised version of the manuscript that addresses the points raised during the review process.

While the compounds is interesting and potentially important, the reviewers pointed out some deficits.

In reviewing their comments and the manuscript, it was felt that there were indeed some gaps in the information that would be required for rigor of the manuscript. These have been indicated in the editor's comments and in the comments from Reviewer 1. 

We look forward to receiving your revised manuscript.

Kind regards,

Noreen J. Hickok, Ph.D.

Academic Editor

PLOS ONE

Additional Editor Comments:

Lines 98-106: It is noted that several authors have interest in a company marketing CZs. This paragraph does not achieve normal scientific equipoise and I would ask that you delete “unique” from “unique first in class” the “More specifically…” sentence to regain that.

Please specify the source of the hyaluoronic acid—based on the web site, it might be bacterial in origin, although that is not clear.

Please add a cartoon of the synthetic scheme with structures, with the CZ-01179 structure clearly identified.

Cytotoxicity methods and Results: Please give us more information about what was done, the timing, cell concentrations, what the scale means (not just pass and fail but percent viability and how it was measured) and number of replicates. The reference to GMP regulations is not sufficient. The results are presented without error bars which leaves us in the dark as to how cytotoxic. Also, the score of 4 leaves no room for nuance. Did it cause 20% toxicity? 80%? The paragraph implies equivalence without information as to that huge range that occupies the failing category. Also, the cytotoxicity assay was performed with 2% gel, but topical gels are 0.5%, 1% and 2%. Some information about the dose dependency of the cytotoxicity is necessary.

We know that most antibiotics require ~1000X the MIC to eradicate biofilm bacteria and for the standard treatments shown, we know the MICs and where the formulations fall. What is the MIC for CZ-01179 for MRSA and for P. aeruginosa?

Figure 6: Please define what the bars represent—presumably MRSA and P. aeruginosa (purple and green) but the text on the gray bars is not readable under normal magnification and needs to be also delineated for MRSA and PA. Also, why is 10^2 CFU/ml considered to be not detectable?

Statistical analysis needs to be performed and the number of samples, as well as the test needs to be specified-

Journal Requirements:

2. Thank you for your comment stating that Figure 1 was used in a previous PLOS ONE publication. PLOS applies the Creative Commons Attribution (CC BY) license to articles and other works we publish. Under this Open Access license, authors agree that anyone can reuse their article in whole or part for any purpose, for free, even for commercial purposes. Anyone may copy, distribute, or reuse the content as long as the author and original source are properly cited. As such, please cite the paper that the image originally appeared in. This may be done in the figure legend, for example, by stating "This figure was reused from [citation] under the Creative Commons Attribution (CC BY) license." Please feel free to reach out to Susan Hepp at shepp@plos.org with any questions.

3. In your Methods section, please provide additional details regarding the assays done by Nelson Laboratories. First, please provide a more detailed description of methods used by the laboratory. In addition, please include the source from which they obtained the cells, the catalog number if applicable, whether the cell line was verified and checked for contamination, and if so, how. For more information on PLOS ONE's guidelines for research using cell lines, see https://journals.plos.org/plosone/s/submission-guidelines#loc-cell-lines.

4. Please report your p-values for the remaining log10 CFU/collagen coupon following 24 h of topical treatment (for control vs. experimental groups in Table 1).

5. We note that you refer to data that is not included in your manuscript in lines 297 and 465. PLOS ONE does not permit references to “data not shown” as per our data availability policy (https://journals.plos.org/plosone/s/data-availability#loc-minimal-data-set-definition). Please either provide this data or remove these lines from the manuscript if they are not integral to the manuscript.

6. Thank you for stating the following in the Competing Interests section:

"I have read the journal's policy and the authors of this manuscript have the following competing interests: DLW, TJH, PHS and REL have financial interest in Curza Global, LLC which licensed the CZ technology from the University of Utah."

We note that one or more of the authors are employed by a commercial company: Curza Global, LLC.

6.1. Please provide an amended Funding Statement declaring this commercial affiliation, as well as a statement regarding the Role of Funders in your study. If the funding organization did not play a role in the study design, data collection and analysis, decision to publish, or preparation of the manuscript and only provided financial support in the form of authors' salaries and/or research materials, please review your statements relating to the author contributions, and ensure you have specifically and accurately indicated the role(s) that these authors had in your study. You can update author roles in the Author Contributions section of the online submission form.

6.2. Please also provide an updated Competing Interests Statement declaring this commercial affiliation along with any other relevant declarations relating to employment, consultancy, patents, products in development, or marketed products, etc. 

Reviewers' comments:

Reviewer's Responses to Questions

**Comments to the Author**

1. Is the manuscript technically sound, and do the data support the conclusions?

Reviewer #1: No

Reviewer #2: Yes

2. Has the statistical analysis been performed appropriately and rigorously? 

Reviewer #1: No

Reviewer #2: Yes

3. Have the authors made all data underlying the findings in their manuscript fully available?

Reviewer #1: Yes

Reviewer #2: Yes

4. Is the manuscript presented in an intelligible fashion and written in standard English?

Reviewer #1: Yes

Reviewer #2: Yes

5. Review Comments to the Author

Reviewer #1: In this article, the authors conduct two tests of a candidate antibiofilm compound for his efficacy against lab grown MRSA and Pseudomonas aeruginosa biofilms. While the need for new biofilm-specific antimicrobial compounds is described by the authors, the introduction is overly long. Much of the biofilm work cited is covered in a number of recent reviews, and I would recommend that the authors focus on review articles specifically mentioning wound biofilms, since their product is targeted towards that end. In looking at the data carefully, there are a number of issues that are of a concern and from this reviewer's perspective the study looks premature in its present form. Specific issues are mentioned below:

1) In the first instance, most wound infections are polymicrobial, yet the authors largely explore monocultures that have been grown on lab media. It would be worthwhile exploring a potential wound biofilm treatment in a mixed community and investigate whether the compound targeted specific members of that population.

2) Given the importance of ESKAPE pathogens, it would be worthwhile to investigate whether the test compound works against more than two members of that group. Certainly those studies could be done in monoculture, then mixed culture experiments could follow.

3) The authors only tested a small number of concentrations at a single time point (48 h). It would really be useful to calculate the MIC or equivalent for biofilm inactivation and also determine whether biofilm age played any role in this.

4) How relevant to the actual clinical situation is the media used for biofilm growth?

5) In at least one figure (Fig 6) the bar graphs would be difficult to read for someone who may be color blind. As well the bars are not defined in the figure legend.

6) In the table 1, the authors report means and standard deviations but no statistical analysis is evident.

7) What was the basis used to add color to the SEM micrographs?

Reviewer #2: The manuscript, "Examination of a first-in-class bis-dialkylnorspermidine-terphenyl antibiotic in topical formulation against mono and polymicrobial biofilms" describes the evaluation of a gel based on CZ-01179 for eradication of mono- and poly-microbial biofilms of MRSA and P. aeruginosa. The authors show convincingly that the CZ-01179 gel works as well or better than current clinical standard topical treatments at killing MRSA and P. aeruginosa biofilms.

This was an elegantly written manuscript and the data was clear, rigorous, and convincing. I honestly could not find any issues with their methodology, their conclusions, or the writing. The only suggestion I have is to include a figure with the structure of CZ-01179 so that the readers don't have to look up the prior publication if they are interested. Excellent job!

6. PLOS authors have the option to publish the peer review history of their article (what does this mean?). If published, this will include your full peer review and any attached files.

Reviewer #1: No

Reviewer #2: No

---

## [Author Response · Author response to Decision Letter 0]

15 Sep 2020

Re: PLOS ONE Decision: Revision require [PONE-D-20-16178] 8/3/2020

Dear Reviewers, 

Thank you for your time to review, your comments and recommendations. They are well-received. We provided responses below.

Additional Editor Comments:

Comment: Lines 98-106: It is noted that several authors have interest in a company marketing CZs. This paragraph does not achieve normal scientific equipoise and I would ask that you delete “unique” from “unique first in class” the “More specifically…” sentence to regain that.

Response: We removed “unique” in all instances and left “first in class.”

Comment: Please specify the source of the hyaluoronic acid—based on the web site, it might be bacterial in origin, although that is not clear.

Response: We called the company and provided updated information in the methods/supplies and reagents: “This HA is a bacterial fermentation product of Streptococcus pyogenes.”

Comment: Please add a cartoon of the synthetic scheme with structures, with the CZ-01179 structure clearly identified.

Response: Great suggestion. This is now Figure 2 and is placed in the section describing synthesis.

Comment: Cytotoxicity methods and Results: Please give us more information about what was done, the timing, cell concentrations, what the scale means (not just pass and fail but percent viability and how it was measured) and number of replicates. The reference to GMP regulations is not sufficient. The results are presented without error bars which leaves us in the dark as to how cytotoxic. Also, the score of 4 leaves no room for nuance. Did it cause 20% toxicity? 80%? The paragraph implies equivalence without information as to that huge range that occupies the failing category. 

Response: Great feedback, thank you. We added this into the methods:

Test articles and controls were extracted in 1x minimal essential media (MEM) with 5% bovine serum for 24-25 h at 37 ± 1° C with agitation. Multiple well cell culture plates were seeded with a verified quantity of industry standard L-929 cells (ATCC CCL-1) and incubated until ~80% confluent. The test articles were held at room temperature for less than four h before testing. The extract fluids were not filtered, centrifuged or manipulated in any way following the extraction process. The test extracts were added to the cell monolayers in triplicate. The cells were incubated at 37 ± 1° C with 5 ± 1% CO2 for 48 ± 3 h. 

Cell monolayers were examined and scored (0-4) based on the degree of cellular destruction. Specifically, Grade 0 = No reactivity, no cell lysis; Grade 1 = Slight reactivity, ≤20% rounding, occasional lysis; Grade 2 = Mild reactivity, 20% ≤ 50% rounding, no extensive cell lysis; Grade 3 = Moderate reactivity, 50% ≤ 70% rounding and lysed cells; Grade 4 = Severe reactivity, nearly complete destruction of cell layers. Testing was performed in compliance with US FDA goods and manufacturing practice (GMP) regulations 21 CFR Parts 210, 211 and 820.

We also added this into the cytoxocitiy results, “…all n=3 samples of clinically-relevant topicals scored a 4 on a scale of 0-4 (score of 3-4 being considered failure and 0-2 considered passing). All n=3 samples of CZ-01179 also had a score of 4. Neosporin® was the only topical product to receive a passing score with scores of 1, 2, and 1 for each of the three samples tested.”

Comment: Also, the cytotoxicity assay was performed with 2% gel, but topical gels are 0.5%, 1% and 2%. Some information about the dose dependency of the cytotoxicity is necessary.

Response: We could have these tests rerun to include more dilute concentrations, but we used the upper end of concentrations as these are the concentrations at which products are used in pre-clinical models and in clinic. We feel this is sufficient for the purposes of the study, but please advise if you’d like us to include more dilute data.

Comment: We know that most antibiotics require ~1000X the MIC to eradicate biofilm bacteria and for the standard treatments shown, we know the MICs and where the formulations fall. What is the MIC for CZ-01179 for MRSA and for P. aeruginosa?

Response: Great question. The MIC of CZ-01179 is ~1-2 μg/mL against both isolates. 

Comment: Figure 6: Please define what the bars represent—presumably MRSA and P. aeruginosa (purple and green) but the text on the gray bars is not readable under normal magnification and needs to be also delineated for MRSA and PA. Also, why is 10^2 CFU/ml considered to be not detectable?

Response: The original Figure 6 is now Figure 7. We reconfigured the figure so it’s more legible and easily readable in grayscale. The bar at 10^2 was misleading. That was just a text box, not demonstrating a number.

Comment: Statistical analysis needs to be performed and the number of samples, as well as the test needs to be specified-

Response: This is all updated in Table 1 and a paragraph discussion in the Discussion section.

Comment: Journal Requirements:

Response: We corrected these items. If we’ve missed something, we’re happy to modify further.

Comment: 2. Thank you for your comment stating that Figure 1 was used in a previous PLOS ONE publication. PLOS applies the Creative Commons Attribution (CC BY) license to articles and other works we publish. Under this Open Access license, authors agree that anyone can reuse their article in whole or part for any purpose, for free, even for commercial purposes. Anyone may copy, distribute, or reuse the content as long as the author and original source are properly cited. As such, please cite the paper that the image originally appeared in. This may be done in the figure legend, for example, by stating "This figure was reused from [citation] under the Creative Commons Attribution (CC BY) license." Please feel free to reach out to Susan Hepp at shepp@plos.org with any questions.

Response: Perfect feedback, thank you! We included the reference and statement in the figure legend.

Comment: 3. In your Methods section, please provide additional details regarding the assays done by Nelson Laboratories. First, please provide a more detailed description of methods used by the laboratory. In addition, please include the source from which they obtained the cells, the catalog number if applicable, whether the cell line was verified and checked for contamination, and if so, how. For more information on PLOS ONE's guidelines for research using cell lines, see https://journals.plos.org/plosone/s/submission-guidelines#loc-cell-lines.

Response: We included additional info from Nelson Labs including the ATCC strain they use. We think it’s sufficient, but let us know if you feel more is needed.

Comment: 4. Please report your p-values for the remaining log10 CFU/collagen coupon following 24 h of topical treatment (for control vs. experimental groups in Table 1).

Response: Thank you. We updated Table 1 with these values, and included a paragraph in the Discussion to correlate statistical significance with clinical significance.

Comment: 5. We note that you refer to data that is not included in your manuscript in lines 297 and 465. PLOS ONE does not permit references to “data not shown” as per our data availability policy (https://journals.plos.org/plosone/s/data-availability#loc-minimal-data-set-definition). Please either provide this data or remove these lines from the manuscript if they are not integral to the manuscript.

Response: We can’t wait to share this data! But it will wait and we removed the sentence.

Comment: 6. Thank you for stating the following in the Competing Interests section:

"I have read the journal's policy and the authors of this manuscript have the following competing interests: DLW, TJH, PHS and REL have financial interest in Curza Global, LLC which licensed the CZ technology from the University of Utah."

We note that one or more of the authors are employed by a commercial company: Curza Global, LLC.

6.1. Please provide an amended Funding Statement declaring this commercial affiliation, as well as a statement regarding the Role of Funders in your study. If the funding organization did not play a role in the study design, data collection and analysis, decision to publish, or preparation of the manuscript and only provided financial support in the form of authors' salaries and/or research materials, please review your statements relating to the author contributions, and ensure you have specifically and accurately indicated the role(s) that these authors had in your study. You can update author roles in the Author Contributions section of the online submission form.

6.2. Please also provide an updated Competing Interests Statement declaring this commercial affiliation along with any other relevant declarations relating to employment, consultancy, patents, products in development, or marketed products, etc. 

Response: We updated financial conflicts in the cover letter.

Comment: 7. Please include captions for your Supporting Information files at the end of your manuscript, and update any in-text citations to match accordingly. Please see our Supporting Information guidelines for more information: http://journals.plos.org/plosone/s/supporting-information.

Response: We updated this section.

Reviewers' comments:

Reviewer's Responses to Questions

Comments to the Author

Comment: Reviewer #1: In this article, the authors conduct two tests of a candidate antibiofilm compound for his efficacy against lab grown MRSA and Pseudomonas aeruginosa biofilms. While the need for new biofilm-specific antimicrobial compounds is described by the authors, the introduction is overly long. Much of the biofilm work cited is covered in a number of recent reviews, and I would recommend that the authors focus on review articles specifically mentioning wound biofilms, since their product is targeted towards that end. In looking at the data carefully, there are a number of issues that are of a concern and from this reviewer's perspective the study looks premature in its present form.

Response: Depending on what the editorial review thinks, I try to allow my students some liberty in how they lay out and present an Intro, etc. If possible, I’d like to leave it in the current form, but am happy to consider modification if necessary. We cited a recently published article with CZ-01179 against A. baumannii to show advancement of CZ-01179. We don’t feel this paper is overly premature, but setting a stage for a platform of data collection.

Specific issues are mentioned below:

Comment: 1) In the first instance, most wound infections are polymicrobial, yet the authors largely explore monocultures that have been grown on lab media. It would be worthwhile exploring a potential wound biofilm treatment in a mixed community and investigate whether the compound targeted specific members of that population.

Response: We agree. This is a direction we’re going with work that is currently ongoing.

Comment: 2) Given the importance of ESKAPE pathogens, it would be worthwhile to investigate whether the test compound works against more than two members of that group. Certainly those studies could be done in monoculture, then mixed culture experiments could follow.

Response: We are collecting these data currently. We also cited our recently published paper with CZ-01179 against A. baumannii.

Comment: 3) The authors only tested a small number of concentrations at a single time point (48 h). It would really be useful to calculate the MIC or equivalent for biofilm inactivation and also determine whether biofilm age played any role in this.

Response: We have the MICs; ~1-2 μg/mL. Given the length of the manuscript, would you like use to include the method for performing MIC, or just include the MIC with a reference to the CLSI standard used for determination? We don’t have age data, but could be addressed in the future. Great suggestion.

Comment: 4) How relevant to the actual clinical situation is the media used for biofilm growth?

Response: BHI is considered closer to clinical relevance than a broth such as TSB. BHI is composed of animal tissues, which may retain at least some similar properties to which organisms are exposed in vivo. We use this for animal work and observe strong signals of infection.

Comment: 5) In at least one figure (Fig 6) the bar graphs would be difficult to read for someone who may be color blind. As well the bars are not defined in the figure legend.

Response: Figure 6 is now Figure 7. We revamped so it’s more legible. Great suggestion, thank you.

Comment: 6) In the table 1, the authors report means and standard deviations but no statistical analysis is evident.

Response: This is now included.

Comment: 7) What was the basis used to add color to the SEM micrographs?

Response: We like the reader to be able to distinguish components more easily.

Reviewer #2: 

Comment: The manuscript, "Examination of a first-in-class bis-dialkylnorspermidine-terphenyl antibiotic in topical formulation against mono and polymicrobial biofilms" describes the evaluation of a gel based on CZ-01179 for eradication of mono- and poly-microbial biofilms of MRSA and P. aeruginosa. The authors show convincingly that the CZ-01179 gel works as well or better than current clinical standard topical treatments at killing MRSA and P. aeruginosa biofilms.

This was an elegantly written manuscript and the data was clear, rigorous, and convincing. I honestly could not find any issues with their methodology, their conclusions, or the writing. The only suggestion I have is to include a figure with the structure of CZ-01179 so that the readers don't have to look up the prior publication if they are interested. Excellent job!

Response: What a review, thank you. We look forward to the next stage of analysis! We included the structure that is now Figure 2.

---

## [Editor Report · Decision Letter 1]

18 Sep 2020

Examination of a first-in-class bis-dialkylnorspermidine-terphenyl antibiotic in topical formulation against mono and polymicrobial biofilms

PONE-D-20-16178R1

Dear Dr. Williams,

We’re pleased to inform you that your manuscript has been judged scientifically suitable for publication and will be formally accepted for publication once it meets all outstanding technical requirements.

Kind regards,

Noreen J. Hickok, Ph.D.

Academic Editor

PLOS ONE

---

## [Editor Report · Acceptance letter]

6 Oct 2020

PONE-D-20-16178R1 

Examination of a first-in-class bis-dialkylnorspermidine-terphenyl antibiotic in topical formulation against mono and polymicrobial biofilms 

Dear Dr. Williams:

I'm pleased to inform you that your manuscript has been deemed suitable for publication in PLOS ONE. Congratulations! Your manuscript is now with our production department. 

Kind regards, 

on behalf of

Dr. Noreen J. Hickok 

Academic Editor

PLOS ONE